# Precision of a Hand-Held 3D Surface Scanner in Dry and Wet Skeletal Surfaces: An Ex Vivo Study

**DOI:** 10.3390/diagnostics12092251

**Published:** 2022-09-18

**Authors:** Jannis Probst, Konstantinos Dritsas, Demetrios Halazonetis, Yijin Ren, Christos Katsaros, Nikolaos Gkantidis

**Affiliations:** 1Department of Orthodontics and Dentofacial Orthopedics, School of Dental Medicine, University of Bern, CH-3010 Bern, Switzerland; 2Department of Orthodontics, School of Dentistry, National and Kapodistrian University of Athens, GR-11527 Athens, Greece; 3Department of Orthodontics, W.J. Kolff Institute, University Medical Center Groningen, University of Groningen, 9700 RB Groningen, The Netherlands

**Keywords:** diagnosis, documentation, computer-assisted, imaging, three-dimensional, digital image processing, bone and bones

## Abstract

Three-dimensional surface scans of skeletal structures have various clinical and research applications in medicine, anthropology, and other relevant fields. The aim of this study was to test the precision of a widely used hand-held surface scanner and the associated software’s 3D model generation-error in both dry and wet skeletal surfaces. Ten human dry skulls and ten mandibles (dry and wet conditions) were scanned twice with an industrial scanner (Artec Space Spider) by one operator. Following a best-fit superimposition of corresponding surface model pairs, the mean absolute distance (MAD) between them was calculated on ten anatomical regions on the skulls and six on the mandibles. The software’s 3D model generation process was repeated for the same scan of four dry skulls and four mandibles (wet and dry conditions), and the results were compared in a similar manner. The median scanner precision was 31 μm for the skulls and 25 μm for the mandibles in dry conditions, whereas in wet conditions it was slightly lower at 40 μm for the mandibles. The 3D model generation-error was negligible (range: 5–10 μm). The Artec Space Spider scanner exhibits very high precision in the scanning of dry and wet skeletal surfaces.

## 1. Introduction

3D surface imaging with hand-held scanners has been widely used in the latest years for various scientific, as well as industrial, applications [1,2]. The scanners’ favorable price, the ease of usage, and the user-friendlier software may have contributed to its increasing popularity [3]. Such scanners are often used to scan skeletal specimens for different applications [4,5]. For example, one of the main goals of radiologic research is the improvement of image quality while keeping the radiation dose as low as possible to minimize the associated risks [6]. However, ethical considerations make the execution of relevant studies on human subjects impossible unless the patient directly benefits from the acquired images, which is rarely the case when repeated exposures are required. To overcome this limitation, dry human skeletal specimens are commonly used in ex vivo radiographic settings, and the missing soft tissues are often simulated by embedding them in water or similar liquid materials [7]. For various research purposes, these specimens are also digitally transformed to three-dimensional (3D) surface models, either segmented from radiographic data, such as computed tomographies (CT) and cone beam computed tomographies (CBCT), or directly scanned with accurate industrial scanners [8,9]. The latter approach is usually preferable when highly accurate surface models are sought because it does not involve the segmentation error, which is unavoidable when obtaining 3D surface models from radiographic images [10,11]. Three-dimensional surface imaging finds additional applications in forensic medicine, mainly for the documentation of crime scenes and accidents [12], as well as in anthropology for digital documentation and analysis of bony specimens [13,14].

The scanning process with hand-held scanners usually involves the acquisition of multiple smaller surface scans which are subsequently processed (image stitching) through appropriate software, resulting to the complete digital model. Despite the abundance of literature in different scientific fields and industrial applications where hand-held scanners are utilized [15], the reproducibility of the scanning process or possible errors occurring during the generation of skeletal surface models have not yet been extensively investigated. In few cases, user-defined landmarks have been used to assess the accuracy, which introduces additional errors [4,5].

Thus, the aim of this study was to test the reproducibility of 3D skeletal surface models obtained with a widely used hand-held scanner, through the comparison of repeatedly scanned bone specimens in both dry and wet conditions. Human dry skulls were selected for the purposes of this study, and reproducibility was tested on smaller (mandibles) and bigger (skulls) sized objects.

## 2. Materials and Methods

### 2.1. Sample

The study sample consisted of ten human dry skulls and ten human mandibles that were acquired from the Municipal cemetery of Serres, Greece, following the relevant application and authorization acquired from the local authorities. The mandibles did not originally belong to the same subjects, but were selected out of a large available number to match the crania based on an approximate fit of their anatomical form, primarily in width and length (mandibular condyles matched to each glenoid fossa and upper matched to lower dentoalveolar structures). The upper part of the cranium was sliced, exposing the anterior cranial base surface.

### 2.2. Data Acquisition

The bone specimens were initially scanned in dry conditions using the structured-light 3D scanner Artec Space Spider (Artec3D, Luxembourg, Software: Artec Studio 12, Version 12.1.6.16). The scanner was calibrated according to the manufacturer’s instructions. Each specimen was scanned while placed on a rotating table (without being fixed) using various orientations to facilitate proper access of the scanner’s sensors to the target surface (Figure 1).

Both the outer surface and the inner anterior cranial base area of the skulls were scanned, as well as the whole surface of the mandibles. The scanning of the skulls consistently started from the face and was followed by the anterior cranial base, which were the structures of interest in this study, and then the rest of the structures were scanned. Usually, two to five consecutive scanning sessions, each consisting of series of single images, were required to scan an entire skull, and two scanning sessions to scan a mandible. The number of sessions required for each specimen depended on the size, the anatomical form and the times that each specimen had to be repositioned on the rotating table for the proper scanning of all of the target surfaces. During surface image acquisition, the surface scanner was connected to a high-end computer (Acer Predator 17 X GX-792-76PB, 17.30”, UHD, Intel Core i7-7700HQ, 32GB, HDD, SSD), which stored all captured images real time. The original data obtained from the scanner were visualized real time and saved through the Artec Studio 12 software in the form of open projects. All dry skulls and mandibles were scanned once more, resulting in another 10 dry skull and 10 dry mandible scans. Thus, two scans were then available for each specimen in dry conditions.

In a second step, 10 mandibles were embedded in water for ten minutes, and were afterwards scanned again shortly after removal. This embedding time was chosen so that the bone specimens were effectively hydrated, while avoiding the permanent damage that could be caused by an extended hydration period. To reduce potential artifacts created by water droplets remaining on the skeletal surfaces, all specimens were placed on a table covered with paper towels and were afterwards briefly patted with them prior to scanning. Following the aforementioned process, each scanning started about 1 min after the removal of the respective specimen from water. The whole process was repeated a few days after the first scan of the mandibles in wet conditions, resulting in another 10 mandible scans. Therefore, two scans were then available for each specimen in wet conditions.

One trained operator (NG) performed all scans applying the same protocol, which was defined according to the manufacturer’s instructions and following pilot testing. The scans were performed within a one-month period in standardized conditions in a room with normal room temperature (22–25 °C) and natural, as well as artificial, ambient light. The total scanning time required for each specimen was approximately 7 min for a complete skull and 2 min for a mandible in dry conditions. The scanning time was increased by approximately 30% in wet surfaces.

### 2.3. Data Post-Processing (3D Model Generation)

The individual images included in the multiple separate scanning sessions of each specimen were manually post-processed in the Artec Studio 16 software (Version 16.0.5.114, Luxembourg, Luxembourg) by a trained operator (JP), to combine the smaller partial scans into a single complete 3D model. The software offers the option of automatic reconstruction of the final surface model using the originally captured individual images. However, certain images showed increased errors, as calculated by the software based on their differences from other images that included common areas. The previous experience of our team with this scanner [16], as well as pilot testing on the current data, showed that a more reliable 3D surface model is obtained when all steps of data processing required to produce the final surface model are controlled manually. The operator has to check the error of individual images, assess the result of each image stitching process required to produce the final model, and discard any images of increased error and any intermediate superimposition processes that do not produce satisfactory outcomes.

More specifically, the auto-alignment feature was used for the initial alignment of the partial scans. In cases where at least one scan was not properly aligned, the scans were aligned semi-automatically through the definition of common landmarks for an initial rough approximation, before the automatic alignment was implemented. Non-essential data, such as artifacts and irrelevant hard tissues away from the targeted anatomical structure of each scan, were manually removed to facilitate the superimposition process, targeting a maximum error of 0.3 mm among all scans. For the final registration of the scans, the rough serial registration function was applied using only geometry, followed by the global registration function at a key frame ratio of 0.3. Before the fusion of the scans into a single complete model, the outlier removal function was applied with the following settings: 3D-noise level 3, 3D resolution 0.3 mm. Finally, the sharp fusion function was applied at a resolution of 0.3 mm, without filling holes. The generated 3D models were post-processed for further removal of possible artifacts by applying a small-object filter, where all objects, except for the largest one, were removed. The final 3D models were then exported as STL files. The post-processing time to create a final 3D surface model through the aforementioned process was approximately 40 min for one skull and 25 min for one mandible.

To assess the 3D model generation error attributed only to data post-scan processing though the Artec Studio 16 software, the above-described post-processing was repeated by the same operator for the scans of four randomly selected skulls in dry conditions and four mandibles in both dry and wet conditions, two weeks after the original processing.

### 2.4. Scanner Precision and 3D Model Generation Error

Each pair of final corresponding 3D surface models exported from the Artec Studio 16 software was imported and superimposed in Viewbox 4 (Version 4.1.0.12, dHAL Software, Kifissia, Greece) using the software’s implementation of the iterative closest point algorithm (ICP) [17]. The following settings were used: 100% estimated overlap, matching point to plane, exact nearest neighbor search, 100% point sampling, and 50 iterations. The superimposition reference areas used for each skull and mandible are depicted in Figure 2. 

Ten measurement areas were defined in each skull, consisting of 10,000 triangles each. These were located at the middle, the right, and the left side of the forehead, at the right and left side of the zygomatic process, at the right and left side of the maxillary complex, at middle part of the sphenoid bone, and the right and left sides of the anterior cranial base (Figure 3).

Six measurement areas were defined in each mandible, consisting of 10,000 triangles each. They were located at the middle part of the mandibular body and at the center of the right and left ramus, buccally and lingually (Figure 3).

The scanner precision was calculated as follows: the two repeated scans of each specimen were approximated using a best-fit algorithm, which was applied on the reference areas shown in Figure 2. Afterwards, the mean absolute distances (MAD) between each two corresponding surface models that were best fit approximated were calculated for each selected circular area shown in Figure 3. Zero MAD would show perfect congruence between the models, and thus perfect reproducibility, that diminishes as MAD is increasing. Qualitative comparisons of the entire models are also presented as color-coded distance maps, allowing for the visualization of possible differences at the original anatomical areas of each specimen. Signed values are shown in the respective images to allow for a more comprehensive outcome assessment. The 3D model generation error was assessed in a similar manner, but using the pairs of models that originated from the repeated post-processing of identical scans. 

### 2.5. Statistical Analysis

The statistical analysis was performed using the IBM SPSS statistics for Windows (Version 28.0. Armonk, NY, USA: IBM Corp.). 

Shapiro–Wilk and Kolmogorov–Smirnov tests on the raw data did not always reveal a normal distribution. Thus, non-parametric statistics were applied. The reproducibility of the scanner (precision) and the software model generation error for each measurement area are shown with box plots. Differences in the amount of error among all measured areas were tested through Friedman’s test, followed by Dunn’s test for pairwise comparisons if differences were detected. The *p*-values were adjusted by the Bonferroni correction when multiple tests were performed.

The effect of wetness on the scanner reproducibility and software model generation error was tested in a paired manner with Wilcoxon’s signed-rank test at an alpha level of 0.05.

## 3. Results

### 3.1. Scanner Precision

There was overall high scanner precision considering the skulls in dry conditions (median MAD: 31 μm; IQR: 16; range: 11–101). Scanner precision was affected by the location of the measurement area (Friedman’s test: *p* = 0.032). Pairwise comparisons revealed no specific differences between areas (Dunn’s test: *p* > 0.05).

Similarly, in the mandibles, the precision in dry conditions was high (median: 25 μm; IQR: 21; range: 10–69), but was slightly lower in wet conditions (median: 40 μm; IQR: 32; range: 12–152; *p* < 0.001, Wilcoxon signed-rank test). In this case, the scanner precision was affected by the location of the measurement area only in wet conditions (Friedman’s test: *p* = 0.047). Pairwise comparisons revealed no specific differences between individual areas (Dunn’s test: *p* > 0.05).

The respective box plots showing scanner precision in skulls and mandibles per measurement area are presented in Figure 4. Overall, the precision in the mandibles was comparable to that in the skulls, as indicated by the median error, as well as by the visual assessment of the respective color-coded distance maps between best fit-approximated repeated scans (Figure 5, Figure 6 and Appendix A).

### 3.2. 3D Model Generation Error

The overall model generation error measured across all areas of the skulls scanned in dry conditions was very low (median: 10 μm; IQR: 6; range: 2–25). The generation error was not affected by the measurement area (*p* > 0.05, Friedman’s test).

In the mandibles scanned in dry conditions, the model generation error measured across all areas was again very low (median: 5 μm; IQR: 2; range: 3–9). In wet conditions it was slightly higher, but still very low (median: 6 μm; IQR: 1; range: 5–9, *p* = 0.011, Wilcoxon signed-rank test). The generation error was not affected by the measurement area in either dry or wet conditions (*p* > 0.05, Friedman’s test).

The box plots for the 3D model generation error in skulls and mandibles are presented in Figure 7, and the respective color-coded distance maps regarding all tested specimens are presented in Appendix A.

## 4. Discussion

The precision of the Artec Space Spider scanner was very high, at approximately 30 μm in dry and 40 μm in wet skeletal surfaces. This result is quite satisfactory considering the anatomical form complexity of the human skull. Modern 3D surface scanners, such as the one tested here, can also capture a colored texture of the model and use it in conjunction with the form of the targeted object to align the partial scans during the 3D model generation process. In the present case, despite the skulls having mostly brown color scales, this might have also enhanced the scanner performance. The software’s 3D model generation error was negligible in all cases at approximately 10 μm.

To our knowledge, this is the first study that comprehensively tested the Space Spider scanner on actual skeletal surfaces, and the first study that tested the effect of wetness on the scanning outcome. The study tested precision, and not trueness, because the true form of the specimens would be difficult to obtain in such high detail. However, the fact that precision was high in all different tested scenarios (various anatomical form configurations in different surface conditions), along with the established trueness of the scanner in various other models [16,18,19,20,21], strongly indicates that high accuracy would also be evident in the skeletal surfaces tested here. The high accuracy of this scanner has already been exhibited in previous studies testing different surfaces such as teeth [16], human skin [18,19], human fingerprints depicted in plastic materials [20], or even in difficult-to-scan objects [21]. So far, the scanner has mostly been used to scan smaller objects or limited surfaces. The human skull approaches the upper limit of the scanner’s capacity in terms of volume capture zone (2000 cm^3^), which amounts to a volume of approximately 1800 cm^3^ [22]. Even close to this limit, the scanner’s performance was quite encouraging, allowing for the scanning of complex structures, such as the cavities of a human skull, with the ease of use of a hand-held scanner.

The scanner performance in wet surfaces is important for several applications, including direct scans of skeletal surfaces during live surgeries, forensic applications where exposed skeletal surfaces might be wet due to bleeding or other reasons, or industrial applications where wet surfaces should be scanned at this status. In wet conditions, both the precision of the scanner and the reproducibility of the software model generation showed a reduction, albeit of small amount. A possible cause might be the retained water droplets in the pores on the surface of the specimens, which increase the light scattering during scan acquisition compared with a dry surface. As a result, the sensor of the scanner registers artifacts, which are false particles away from the targeted surface. The increased number of artifacts during the partial scan alignment negatively affects the 3D registration process and leads to inaccuracies [23].

Both skulls and mandibles presented similar reproducibility despite morphological differences in their skeletal configuration. Direct statistical testing was not performed in this case due to the differences in the extent of the measurement surfaces of the skulls compared with that of the mandibles, which could skew the outcomes. However, this concerns only the measurement areas used for the quantitative analysis. When assessing the entire color-coded distance maps, it became apparent that the skulls had worse reproducibility in areas distanced from the superimposition references, such as the temporal bone. This is an expected finding considering that the corresponding models were best-fit approximated at a predefined superimposition reference area, and thus any small rotations of the models on this area results in larger rotations, and thus higher distances, on distant areas, with the center of rotation located somewhere at the center of the selected reference area. This argument is also supported by the outcomes of previous craniofacial superimposition studies testing different hypotheses [11,24,25,26,27].

This study focused on facial and on anterior cranial base skeletal surfaces. Both the superimposition reference, areas as well as the measurement areas, were located on these anatomical structures for reasons described above. We selected the specific areas in the skull as the most relevant for facial outcomes that are evenly distributed over the area of interest to allow for thorough testing of the scanner performance. The facial structures were selected because of their high clinical relevance and importance for human life [28], as well as because of their complexity in form. The anterior cranial base structures were selected because this area is often used as superimposition reference when comparing consecutive patient scans [29,30], and also because its anatomy is totally different than that of the facial hard tissues. Mandibles were also assessed representing facial structures that might have more resemblance to different skeletal configurations, such as the long bones. Thus, the scanner performance was tested in different conditions, and the outcomes could be generalized for broader application.

The measurement area did not affect the scanner precision to a substantial degree. Overall, statistically significant differences were detected, but they were of a small magnitude, and no specific difference was evident when pairwise comparisons between different areas were performed. This could partially be attributed to the correction of the level of significance to avoid false positive results which, however, negatively affects the power of the applied tests. In short, the outcomes were relatively consistent between areas and between specimens, and the detected differences were of small magnitude, allowing for safe conclusions to be drawn.

A limitation of this study was the fact that only one operator performed the scanning procedure. Due to the sensitivity of the scanned objects that required careful handling and time restrictions related to the large amount of generated data, it was decided not to enlist a second operator. Therefore the operator-dependent error could not be assessed. However, the 3D model generation software is mostly automated, and the operator performed the scans following relevant training and according to manufacturer’s instructions. Thus, under these circumstances we do not expect large deviations if another operator was involved. The error of the Viewbox 4 software in processing surface models is negligible, as verified in previous studies testing various anatomical surface models [10,11,16,23]. Another limitation could be that the present study comprehensively tested precision through repeated measurements, but not accuracy or trueness. However, as previously stated, the high accuracy of the specific scanner is well established for different objects and surfaces [16,18,19,20,21,22] and, thus, it is also expected to be evident in skeletal surfaces as the ones tested here.

Important inferences of the present study are that such scans can be used to produce accurate digital 3D models of bone specimens for radiologic research, forensic applications, in medicine (e.g., inter-operation scans), or for anthropological data. These models allow for the acquisition of gold-standard measurements in ex vivo studies that can be used to evaluate the accuracy of most commonly applied imaging techniques, such as the computed tomography (CT) and the cone bean CT (CBCT), which are more error prone, especially when low-radiation protocols are applied for radiation protection issues [8].

## 5. Conclusions

The Artec Space Spider scanner exhibited very high precision in the scanning of dry and wet craniofacial skeletal specimens. The associated software’s 3D model generation error was negligible. Thus, the present scanner is suggested for relevant medical, forensic, anthropological, or other applications where high precision levels are targeted.

## Figures and Tables

**Figure 1 diagnostics-12-02251-f001:**
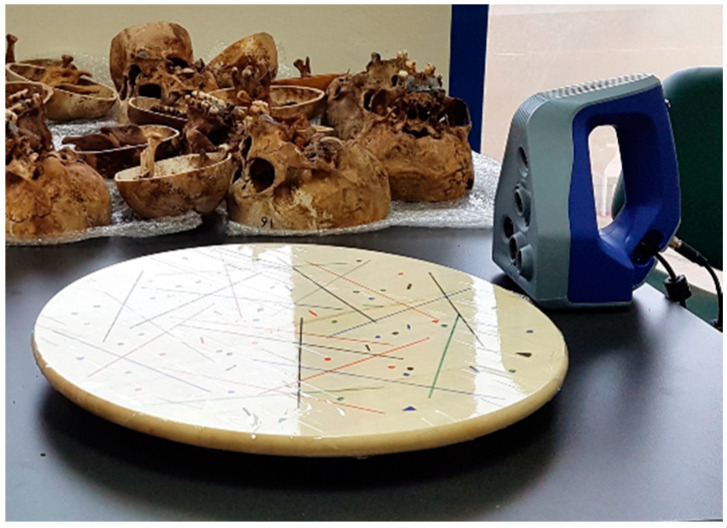
The Artec Space Spider scanner, the rotating table used to perform the scans and selected specimens.

**Figure 2 diagnostics-12-02251-f002:**
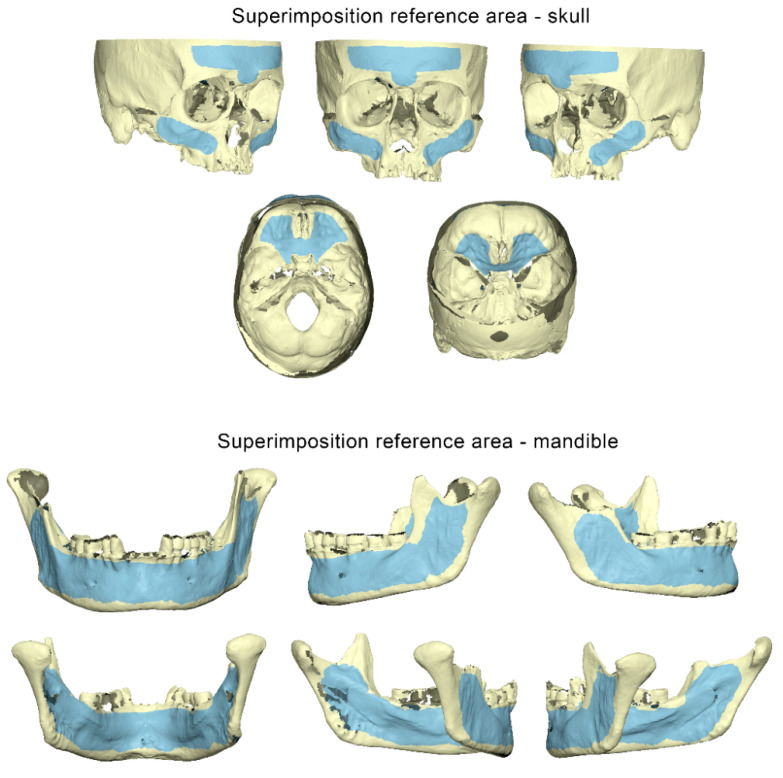
Superimposition reference areas used in the skulls and mandibles (**top**: buccal aspect, **bottom**: lingual aspect) to register corresponding 3D surface models, depicted in light blue color.

**Figure 3 diagnostics-12-02251-f003:**
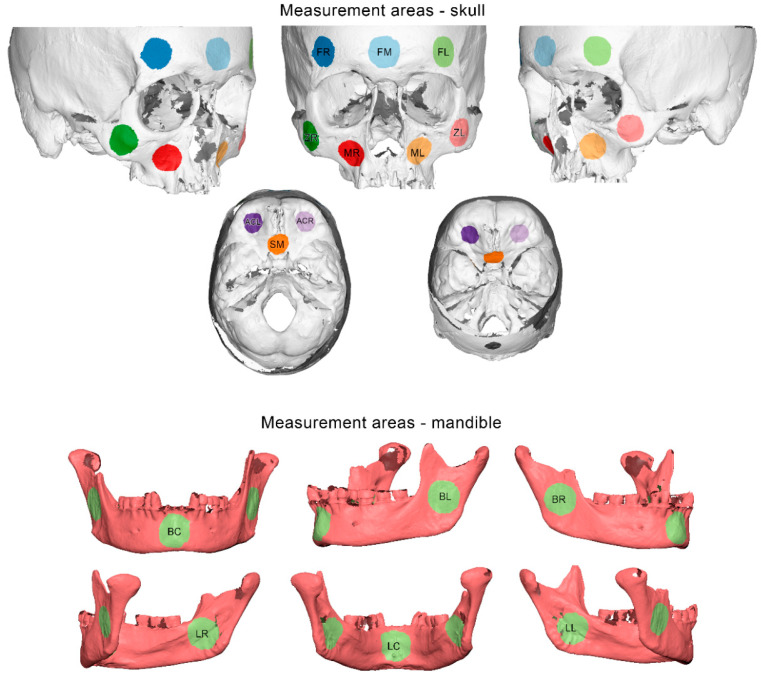
Measurement areas used to assess differences between corresponding 3D surface models shown as colored round surfaces. Skull: forehead left (FL), right (FR), and middle (FM), maxilla left (ML) and right (MR), zygomatic bone left (ZL) and right (ZR), anterior cranial base left (ACL) and right (ACR), and sphenoid bone middle (SM). Mandible: upper row: buccal left (BL), right (BR), and center (BC). Bottom row: lingual left (LL), right (LR), and center (LC).

**Figure 4 diagnostics-12-02251-f004:**
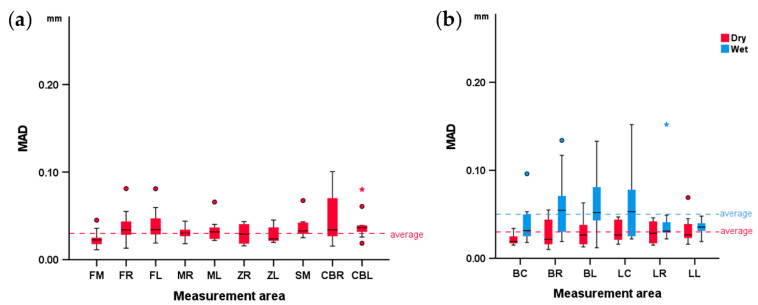
Box plots showing the scanner reproducibility at each measurement area in (**a**) dry conditions at the skulls and (**b**) in dry and wet conditions at the mandibles. MAD: mean absolute distance. Skull: forehead left (FL), right (FR), and middle (FM), maxilla left (ML) and right (MR), zygomatic bone left (ZL) and right (ZR), anterior cranial base left (ACL) and right (ACR), and sphenoid bone middle (SM). Mandible: buccal left (BL), right (BR), and center (BC) and lingual left (LL), right (LR), and center (LC).

**Figure 5 diagnostics-12-02251-f005:**
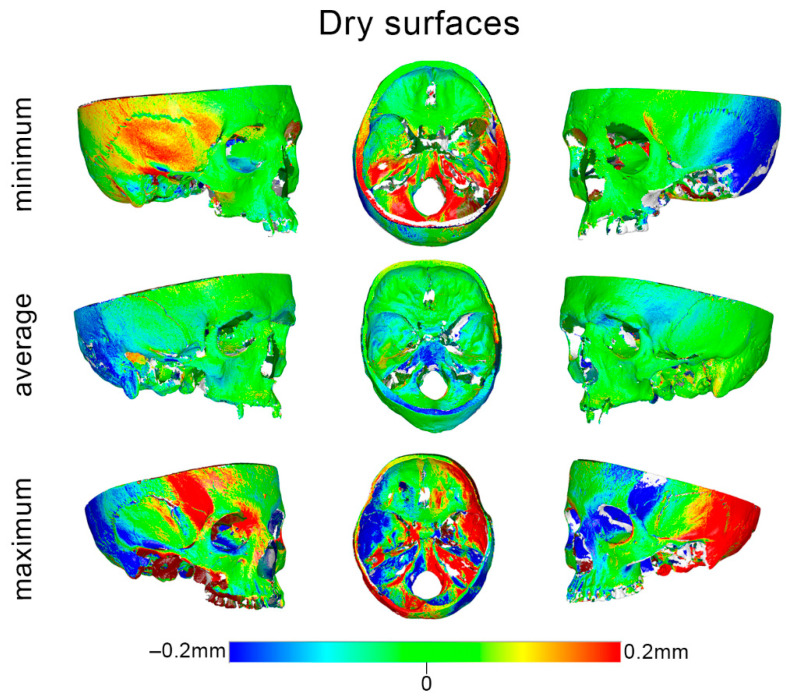
Color-coded distance maps of skulls showing the scanner reproducibility, depicted through selected models that corresponded to the minimum, average, and maximum error detected in the pre-specified measurement areas. Zero distance between best-fit-approximated repeated scans indicates perfect reproducibility.

**Figure 6 diagnostics-12-02251-f006:**
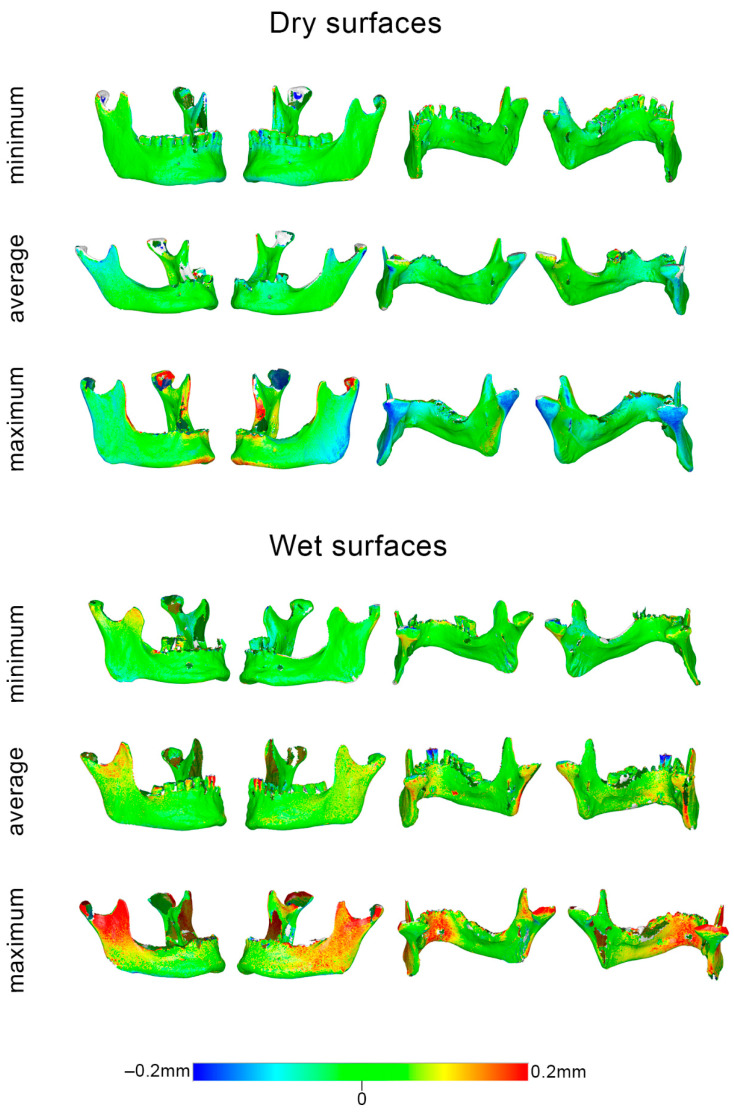
Color-coded distance maps of mandibles showing the scanner reproducibility in dry and wet conditions, depicted through models that corresponded to the minimum, average, and maximum error detected in the pre-specified measurement areas. Zero distance between best-fit-approximated repeated scans indicates perfect reproducibility.

**Figure 7 diagnostics-12-02251-f007:**
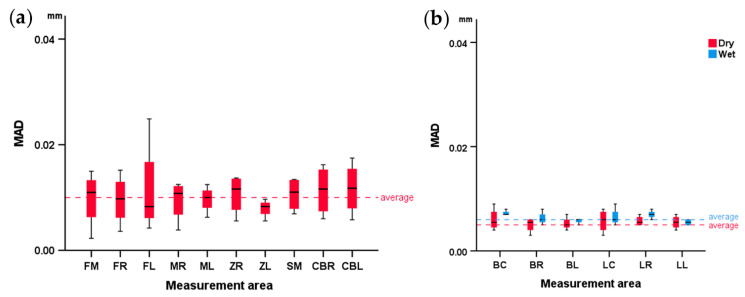
Box plots showing the software’s 3D model generation error at each measurement area in (**a**) dry conditions at the skulls and (**b**) dry and wet conditions at the mandibles. MAD: mean absolute distance. Skull: forehead left (FL), right (FR), and middle (FM), maxilla left (ML) and right (MR), zygomatic bone left (ZL) and right (ZR), anterior cranial base left (ACL) and right (ACR), and sphenoid bone middle (SM). Mandible: buccal left (BL), right (BR), and center (BC) and lingual left (LL), right (LR), and center (LC).

## Data Availability

The data presented in this study are available on request from the corresponding author.

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
