# Peer review of "Precision of a Hand-Held 3D Surface Scanner in Dry and Wet Skeletal Surfaces: An Ex Vivo Study"

_diagnostics, 2022, doi:10.3390/diagnostics12092251_

Round 1
Reviewer 1 Report
The topic of this paper is interesting and up-to-date. Here are my comments and recommendations:
1. Since you discuss hand-held scanner it should be emphasized in the title of your paper.
2. Some referencing to relevant literature are missing:
a. https://www.sciencedirect.com/science/article/abs/pii/S0379073805005773
b. https://link.springer.com/article/10.1007/s10278-012-9487-1
c. https://link.springer.com/article/10.1007/s00586-021-06769-5
Authors seem not did their very best in examining the literature on determining accuracy of 3D scanners. This also includes standards.
3. It is hard to determine the precision of your system as it would require to determine the uncertainty of your instrumentation and post-processing. As a reference, more precise measurement system should have been used here. Your study is more about the repeatability of your technique (both hardware and software) and the effect of dry vs. wet condition. This is crucial here and has to be addressed carefully in the revised paper!
4. What is the MPE of your scanner? How does that value correspond to what you receive in your comparison study?
5. The paper is more a technical report than original research paper. The conclusion refers to only that particular scanner you analyzed. The generalization and the importance of your study is greatly limited.
Author Response
Manuscript ID: diagnostics-1824120
Title: Precision of a 3D surface scanner in dry and wet skeletal surfaces: an ex vivo study
Response to Reviewers' comments
We thank the Reviewers and the Editor for their effort in assessing our manuscript. We did our best to address all issues of concern and we provide below a point by point response to each comment. Relevant adjustments have also been performed in the manuscript using track changes mode, so that the revisions can be easily followed.
Reviewer 1
Reviewer’s comment: The topic of this paper is interesting and up-to-date. Here are my comments and recommendations:
Since you discuss hand-held scanner it should be emphasized in the title of your paper.
Authors’ response: We are happy to see that the reviewer found our study interesting. The term hand-held has been added in the title.
Reviewer’s comment: Some referencing to relevant literature are missing:
- https://www.sciencedirect.com/science/article/abs/pii/S0379073805005773
- https://link.springer.com/article/10.1007/s10278-012-9487-1
- https://link.springer.com/article/10.1007/s00586-021-06769-5
Authors seem not did their very best in examining the literature on determining accuracy of 3D scanners. This also includes standards.
Authors’ response: We thank the reviewer for the suggestions. We have added these references in the introduction and the discussion sections. These references were not originally cited because they tested different scanner systems, but we agree with the reviewer that they are relevant.
Reviewer’s comment: It is hard to determine the precision of your system as it would require to determine the uncertainty of your instrumentation and post-processing. As a reference, more precise measurement system should have been used here. Your study is more about the repeatability of your technique (both hardware and software) and the effect of dry vs. wet condition. This is crucial here and has to be addressed carefully in the revised paper!
Authors’ response: As per definition, precision is how close or dispersed are the repeated measurements to each other and does not consider anyhow the true value/condition. The assessment of accuracy or trueness required a reference to the true value and therefore it was not possible to be tested in the present study. We read the manuscript carefully and performed the necessary revisions to clarify this issue. Please see lines 331-338.
Reviewer’s comment: What is the MPE of your scanner? How does that value correspond to what you receive in your comparison study?
Authors’ response: In our study the MPE is not applicable as the expected value (difference between repeated measurements) is always zero (perfect congruence between identical models). For your information, we could not find an estimated MPE in the literature for the specific scanner that we used. A study has reported a MPE of 1.4% for a less precise scanner of the same company (the Artec Eva), but we deemed this information not relevant to our study setup.
Reviewer’s comment: The paper is more a technical report than original research paper. The conclusion refers to only that particular scanner you analyzed. The generalization and the importance of your study is greatly limited.
Authors’ response: We share the opinion of the reviewer that each device should be subjected to specific testing to be considered reliable for different applications. Therefore, we agree that in such studies generalizability of the present outcomes for other devices or applications is not possible.
Reviewer 2 Report
The purpose of this manuscript is to investigate the precision of a 3D surface scanner on dry and wet human skulls and mandibles ex vivo. The methods and procedures are in detailed described in the manuscript. The information provided in this manuscript is important and can be a nice reference for the readers who are interesting in using 3D surface scanners to scan bones.
The reviewer provides several suggestions for the authors to further improve the manuscript, as below. Thanks and best wishes.
General comments:
1. The data post-processing involves a lot of manual operations. It can be well understood. However, it can be imagined that the precision of a 3D surface scanner may be largely dependent on the techniques of the operator, including the technique to perform the scan (i.e., data acquisition) and the technique to perform the data post-processing. Hence, the technique of the operator is an important factor and variable that may have a profound influence on the precision of a 3D surface scanner. Please discussion more about this issue in the Discussion section (the reviewer acknowledges that you have discussed some as a limitation), and please add some sentences to the Abstract to mention the consideration about this important issue.
2. Please incorporate all the figures and legends in the supplementary material into the main manuscript. The readability could be much better if there is no need for the readers to seek the separate supplementary material for these figures. In the reviewer’s personal opinion, there could be no need to separately present these figures in a supplementary material for this manuscript.
3. The reviewer suggests that the rewriting must be improved. The reviewer cannot well understand the meanings of many sentences throughout the manuscript.
Specific comments:
1. Lines 67 - 70, regarding “The mandibles did not… dentoalveolar structures)”:
First, please rewrite this sentence to improve the readability.
Second, please in detailed describe the criteria for selecting a mandible that is anatomically matched with a cranium. In other words, how to you precisely know a mandible is matched with a cranium?
2. Line 77, regarding “or being repositioned on it”:
The reviewer does not understand this sentence. Please explain.
3. Line 83, regarding “two to five consecutive scanning sessions”, and Lines 84 regarding “… two scanning sessions…”:
Does this number needed to complete the scan depend on the technique of the operator? Please add this information to the manuscript.
4. Lines 89 - 90, regarding “All dry skulls and mandibles were… of repeated scans”:
The reviewer does not understand what this sentence means. Please rewrite this sentence to improve the readability.
5. Line 91, regarding “all bone specimens were embedded in water for ten minutes…”:
How did you know the duration of 10 minutes was a proper duration for the bone to be effectively hydrated? How did you design this duration?
6. Lines 98 - 100, regarding “The whole… scans in wet conditions”:
The reviewer does not understand what this sentence means. Please rewrite this sentence to improve the readability.
7. Lines 139 – 142:
First, please rewrite this sentence to improve the readability.
Second, it is good to know you did this procedure. But, how did you perform the relevant analysis, and where are the relevant results? Please describe.
8. Line 154 and Line 159, regarding “10.000 triangles”:
Does it mean 10 or 10000 triangles? Please clarify.
9. Lines 162 - 164, regarding “The scanner precision was… at each measurement area”:
The reviewer does not understand what this sentence means. Please rewrite this sentence to improve the readability.
Author Response
Reviewer 2
Reviewer’s comment: The purpose of this manuscript is to investigate the precision of a 3D surface scanner on dry and wet human skulls and mandibles ex vivo. The methods and procedures are in detailed described in the manuscript. The information provided in this manuscript is important and can be a nice reference for the readers who are interesting in using 3D surface scanners to scan bones.
The reviewer provides several suggestions for the authors to further improve the manuscript, as below. Thanks and best wishes.
Authors’ response: We are happy to see that the reviewer found our study important and we considered all provided useful suggestions carefully in the revision of our manuscript.
Reviewer’s comment: The data post-processing involves a lot of manual operations. It can be well understood. However, it can be imagined that the precision of a 3D surface scanner may be largely dependent on the techniques of the operator, including the technique to perform the scan (i.e., data acquisition) and the technique to perform the data post-processing. Hence, the technique of the operator is an important factor and variable that may have a profound influence on the precision of a 3D surface scanner. Please discussion more about this issue in the Discussion section (the reviewer acknowledges that you have discussed some as a limitation), and please add some sentences to the Abstract to mention the consideration about this important issue.
Authors’ response: This is a valid point made by the reviewer. We now mention in the abstract that only one operator was involved, and we elaborated more on this issue in the limitations section.
Reviewer’s comment: Please incorporate all the figures and legends in the supplementary material into the main manuscript. The readability could be much better if there is no need for the readers to seek the separate supplementary material for these figures. In the reviewer’s personal opinion, there could be no need to separately present these figures in a supplementary material for this manuscript.
Authors’ response: We agree with the reviewer that the incorporation of supplementary figures in the main text would facilitate proper reading, and therefore, we reorganised the manuscript accordingly. We reduced the number of supplementary figures, but we think that the inclusion of all the supplementary figures in the main text is too expansive, and we prefer to keep certain figures in the supplement.
Reviewer’s comment: The reviewer suggests that the rewriting must be improved. The reviewer cannot well understand the meanings of many sentences throughout the manuscript.
Specific comments: Lines 67 - 70, regarding “The mandibles did not… dentoalveolar structures)”:
First, please rewrite this sentence to improve the readability.
Second, please in detailed describe the criteria for selecting a mandible that is anatomically matched with a cranium. In other words, how to you precisely know a mandible is matched with a cranium?
Authors’ response: The sentence has been rewritten to highlight that the matching of the mandibles was primarily based on their size, namely the width and length. For the purposes of the present study imprecision in matching or the fact that the mandibles did not belong to the same person do not affect anyhow the outcomes.
Reviewer’s comment: Line 77, regarding “or being repositioned on it”:
The reviewer does not understand this sentence. Please explain.
Authors’ response: The sentence has been simplified according to the reviewer’s comments.
Reviewer’s comment: Line 83, regarding “two to five consecutive scanning sessions”, and Lines 84 regarding “… two scanning sessions…”:
Does this number needed to complete the scan depend on the technique of the operator? Please add this information to the manuscript.
Authors’ response: The following sentence has been added in the “2.2. Data acquisition” section: The number of sessions required for each specimen depended on the size, the anatomical form and the times that each specimen had to be repositioned on the rotating table for the proper scanning of all the target surfaces. The issue of the potential operator effects is now discussed further in the limitations section.
Reviewer’s comment: Lines 89 - 90, regarding “All dry skulls and mandibles were… of repeated scans”:
The reviewer does not understand what this sentence means. Please rewrite this sentence to improve the readability.
Authors’ response: The sentence has been rewritten.
Reviewer’s comment: Line 91, regarding “all bone specimens were embedded in water for ten minutes…”:
How did you know the duration of 10 minutes was a proper duration for the bone to be effectively hydrated? How did you design this duration?
Authors’ response: As there is lack of evidence in the literature for this matter, this decision was taken arbitrarily based on what we considered reasonable in order to achieve proper hydration, but permanent damage that could be caused by an extended hydration period. This was reported in the manuscript.
Reviewer’s comment: Lines 98 - 100, regarding “The whole… scans in wet conditions”:
The reviewer does not understand what this sentence means. Please rewrite this sentence to improve the readability.
Authors’ response: The sentence has been rewritten.
Reviewer’s comment: Lines 139 – 142:
First, please rewrite this sentence to improve the readability.
Second, it is good to know you did this procedure. But, how did you perform the relevant analysis, and where are the relevant results? Please describe.
Authors’ response: The analysis and results have been provided in detail in the section ”3.2 3D model generation error.”
Reviewer’s comment: Line 154 and Line 159, regarding “10.000 triangles”:
Does it mean 10 or 10000 triangles? Please clarify.
Authors’ response: Thank you for noticing this. Due to oversight we used the decimal point instead of the comma as a thousand separator. The numbers have now been fixed.
Reviewer’s comment: Lines 162 - 164, regarding “The scanner precision was… at each measurement area”:
The reviewer does not understand what this sentence means. Please rewrite this sentence to improve the readability.
Authors’ response: The sentence has been revised accordingly.
Round 2
Reviewer 1 Report
Thank you for improving your manuscript.
Reviewer 2 Report
The authors have addressed all of the reviewer's comments satisfactorily. The quality of the manuscript has been significantly improved. The reviewer believes that this manuscript can be accepted for publication in Diagnostics. Congratulations and best wishes.